# Expression Patterns of Clock Gene mRNAs and Clock Proteins in Human Psoriatic Skin Samples

**DOI:** 10.3390/ijms23010121

**Published:** 2021-12-23

**Authors:** Viktória Németh, Szabina Horváth, Ágnes Kinyó, Rolland Gyulai, Zsuzsanna Lengyel

**Affiliations:** Department of Dermatology, Venereology and Oncodermatology, Medical School, University of Pécs, H-7632 Pecs, Hungary; nemeth.viktoria@pte.hu (V.N.); horvath.szabina@pte.hu (S.H.); kinyo.agnes@pte.hu (Á.K.); gyulai.rolland@pte.hu (R.G.)

**Keywords:** skin, psoriasis, circadian rhythm

## Abstract

Psoriasis is a systemic inflammatory skin disorder that can be associated with sleep disturbance and negatively influence the daily rhythm. The link between the pathomechanism of psoriasis and the circadian rhythm has been suggested by several previous studies. However, there are insufficient data on altered clock mechanisms in psoriasis to prove these theories. Therefore, we investigated the expression of the core clock genes in human psoriatic lesional and non-lesional skin and in human adult low calcium temperature (HaCaT) keratinocytes after stimulation with pro-inflammatory cytokines. Furthermore, we examined the clock proteins in skin biopsies from psoriatic patients by immunohistochemistry. We found that the clock gene transcripts were elevated in psoriatic lesions, especially in non-lesional psoriatic areas, except for *rev-erbα*, which was consistently downregulated in the psoriatic samples. In addition, the REV-ERBα protein showed a different epidermal distribution in non-lesional skin than in healthy skin. In cytokine-treated HaCaT cells, changes in the amplitude of the *bmal1*, *cry1*, *rev-erbα* and *per1* mRNA oscillation were observed, especially after TNFα stimulation. In conclusion, in our study a perturbation of clock gene transcripts was observed in uninvolved and lesional psoriatic areas compared to healthy skin. These alterations may serve as therapeutic targets and facilitate the development of chronotherapeutic strategies in the future.

## 1. Introduction

The skin is the largest organ of the body, accounting for about 15% of the total body weight in adult humans. It is not surprising that its multiple functions are deeply influenced by other systems such as the autonomic nervous system, hormonal system, metabolism, thermoregulation and immune system. Investigation of the temporal organization of cellular physiology has gained remarkable attention in cellular biology in the last decade. The mammalian circadian (~24 h) timing system generates daily rhythms that are crucial for various physiological processes, such as the sleep–wake cycle, hormone secretion, core body temperature, metabolism and cell cycle control.

The circadian rhythm is controlled by the central regulator, or master clock, which is located in the suprachiasmatic nucleus (SCN). These centrally generated systemic rhythms coordinate molecular clocks in peripheral organs, including the skin [1]. In the last 20 years, several clock genes have been identified and implicated in the molecular regulation of circadian rhythms both in the SCN and in peripheral tissues. The clock mechanisms function by transcription and translation feedback loops of circadian clock genes and their proteins [2]. It has been shown in several studies that the skin also has an intrinsic clock that interacts with central signals. Recent studies have demonstrated that the circadian regulation of skin physiology is under the control of clocks genes [3,4]. Rhythmic patterns in some biophysical and physiological parameters of human skin are also well known (e.g., skin temperature, sebum production) [5,6,7]. Despite these findings, our knowledge about how the circadian system affects specific skin diseases is not complete.

Psoriasis is a chronic, immune-mediated inflammatory skin disease associated with Th17 and also Th1 and Th22 pathways. It is estimated to affect about 2–4% of the population in Western countries [8,9]. Epidemiological studies have shown that the disruption of the circadian system (e.g., shift work) leads to an increased risk of certain skin conditions, such as psoriasis [10]. A recent study using an online survey showed diurnal and seasonal changes in psoriasis. The authors found that psoriatic itch and flares of the disease are more prominent in the evening and during the night. Furthermore, they observed a seasonal pattern of the disease with a trend of flaring during the winter season [11].

On a molecular level, a direct connection was found between Th17 cell differentiation and the circadian clock [12]. Nuclear-factor-interleukin 3 (NFIL3), which regulates a number of immune processes, was shown to suppress Th17 cell development. Nfil3 itself is repressed by BMAL1/CLOCK-dependent expression of REV-ERBα, which shapes the daily rhythm of *nfil3* mRNA contents in CD4^+^ T cells and links to a daily rhythm in Th17 lineage specification [13,14]. Recent findings suggest that *clock* is a novel regulator of psoriasis-like skin inflammation in mice via direct modulation of IL-23R expression in γ/δ+ T cells [15]. It has been also reported that the core clock gene *bmal1* is a negative regulator of the interferon-sensitive gene response in mice, whereby the inflammatory skin condition (psoriasis) is induced by the usage of imiquimod. The authors also showed that skin immune response can be modulated by feeding time [16].

Cytokine production, including the main cytokines in psoriasis inflammation, exhibit diurnal changes. With a nocturnal or early morning peak, serum levels of TNF-α, IFN-γ, IL-1, IL-2, IL-6 and IL-12 oscillate in a circadian manner [17]. Some of these serum levels are significantly elevated in patients with psoriasis [18].

The results of the above mentioned studies highlight a possible influence of the circadian clock system in psoriasis. However, there are insufficient data on human skin samples to know whether clock mechanisms are altered in psoriasis or not. Therefore, to gain more information on whether there is a difference in the circadian transcriptional control, we analyzed the expression of circadian clock genes and the daily rhythm in keratinocytes under inflammatory conditions and in lesional and non-lesional psoriatic skin.

## 2. Results

### 2.1. Altered Clock Gene and Protein Expression Levels in Non-Lesional Psoriatic Skin

Asymptomatic psoriatic skin appears as clinically similar to normal skin, although it can be exposed to the systematic effects of the disorder. Therefore, we compared the oscillation and relative mRNA expression of clock genes between the healthy and psoriatic uninvolved skin samples.

We detected a disturbance of the circadian clock in psoriatic non-lesional skin, affecting nearly all of the genes tested. Under normal circumstances, the *clock* gene does not display a circadian rhythm; however, in non-lesional psoriatic skin, it shows a significant upregulation in the morning samples (Figure 1A). In healthy skin, the quantity of the *bmal1* transcript gradually increases during the day and reaches its maximum at night. In our study, the level of the *bmal1* mRNA decreased in the non-lesional skin biopsies compared to the morning samples and then rose again at midnight (Figure 1B). The amplitude of period genes peaks at noon in the intact skin, while in asymptomatic samples we detected a morning maximum (Figure 1C,D). *Cry1* mRNA is expressed with an afternoon peak in both healthy and non-lesional skin samples, although with higher amplitude in the latter (Figure 1E). The rhythm of *rev-erbα* transcripts with a minimum at noon was detected as unchanged in the two sample types (Figure 1F).

In addition to the disruption of the circadian rhythm, the relative expression of the clock genes in non-lesional psoriatic skin changes significantly relative to the normal skin (Figure 1G). The *cry1*, *bmal1*, *per1* and *per2* genes show very high upregulation in non-lesional skin. In the case of the *clock* transcript, there were significant increases at 06:00 and 00:00 in asymptomatic biopsies, while a similar expression to healthy skin was found at midnight. For *rev-erbα*, we found consequent mild downregulation at all three time points compared to normal skin.

Regarding protein levels, we could not detect a circadian oscillation of the investigated proteins (CLOCK, PER2, CRY1, REV-ERBα) in either sample by immunohistochemistry. The epidermal staining of clock proteins was both cytoplasmic and nuclear (Figure 2A). In the case of the REV-ERBα protein, different epidermal distributions were observed in healthy and non-lesional psoriatic skin. Intense cytoplasmic but no nuclear staining was observed in the healthy epidermis, and the stratum basale was negative with anti-rev-erbα antibody. In contrast, all layers of the epidermis were stained in psoriatic non-lesional samples, and cytoplasmic positivity was accompanied by nuclear staining. There was a significant difference in both nuclear and cytoplasmic staining in the non-lesional epidermis compared to healthy skin (Figure 2B,C). Regarding CRY1, PER2 and CLOCK protein expression, we could not detect differences between the skin biopsies (data not shown).

### 2.2. Circadian Rhythm in Cytokine-Induced Psoriasis Model and in Psoriatic Lesional Skin

We examined whether the key cytokines (TNFα, IL-22, IL-1β, IL-17A) implicated in the pathogenesis of psoriasis modulate the circadian oscillation of the clock genes in HaCaT keratinocytes (Figure 3A–F, Appendix A). These cytokines induce proliferation in the HaCaT cell line in in vitro experiments [19,20]. In our investigations, the circadian genes are expressed as reported by others [3] in untreated keratinocytes; that is, *clock* showed no circadian oscillation at all and the cytokine treatments did not induce substantial alterations in the daily rhythm compared to the control keratinocytes (Figure 3A). Through stimulation with cytokines, the *bmal1* oscillation increased during the cultured period compared to the control. Furthermore, the mRNA levels reached the minimum at 20 h in IL1β and IL-17A treated keratinocytes, 4 h earlier than the controls, IL-22 or TNFα-induced cells, respectively (Figure 3B). The daily rhythm of *rev-erbα* was affected by all cytokines except IL-22. Significantly higher amplitudes were detected at 16–24 h and 36–44 h by TNFα, 12–16 h and 24–44 h by IL-1β and 12–16 h and 32–44 h by IL-17A. The oscillation of the *rev-erbα* transcript peaked 4 h earlier at 16 h, with a significant increase in amplitude by IL-1β and IL-17A compared to the untreated cells (Figure 3F). The cytokine treatment did not affect the daily rhythm of *cry1*, *per1* or *per2* transcripts (Figure 3C–E). Although not statistically significant, the *per1* daily rhythm did increase with IL-17A and IL-22, while *cry1* oscillation was enhanced with TNFα treatment. Alongside the daily rhythm, we examined the relative mRNA expression levels every 4 h after the cytokine treatments and compared them to untreated cells (Figure 3G). Decreases or moderate increases in mRNA expression of the studied clock genes were detected in the IL-22, IL-1β and IL-17A groups. In TNFα-treated HaCaT cells, *clock* mRNA showed significantly increased levels of expression at all times of sampling. Similar—but not statistically significant—results were found for *per2* and *cry1*. Moreover, moderate downregulation of *per1* and *rev-erbα* was observed in all cytokine-treated cells.

The results of the in vitro experiments were compared with data obtained from the symptomatic skin. While changes in the amplitude of oscillation were observed in cytokine-treated HaCaT cells, ex vivo we observed a disruption of the diurnal rhythm in some cases (Figure 1A–F). In lesional psoriatic skin, the diurnal variation of *clock*, *bmal1* and *per2* genes matched the rhythm detected in healthy skin (Figure 1A,B,D). However, *rev-erbα* and *cry1* genes, whose products are thought to have an anti-inflammatory role, showed altered rhythmicity in symptomatic skin and reduced relative mRNA expression compared to healthy skin (Figure 1E,F). Presumably, the complex inflammatory cascade in psoriasis is also responsible for the changes in the circadian system, and no particular component can be singled out. In examining the immunostaining for different epidermal distributions of the analyzed proteins in healthy and psoriatic skin, we detected strong CLOCK- and PER2-positive staining in the stratum granulosum of lesional epidermis samples compared to the healthy and non-lesional psoriatic skin samples (Figure 2A).

The daily rhythms of IL-17A, IL-22 and IL-23 mRNA in symptomatic skin samples were investigated. At the transcript level, cytokine expression levels were elevated at night, decreased at dawn and then increased during the day (Figure 1H).

### 2.3. Expression of Cell Cycle Regulators in Psoriasis

Clock proteins are important transcription factors for cell cycle regulator genes. We investigated the MDM2 and COP1 ubiquitin ligases, which are regulated by members of the circadian loop (PER2 and CRY1, respectively) through negative feedback and may be involved in inflammatory processes [21,22]. As with the clock genes, we examined these two cell cycle regulators in mRNA and protein levels in human skin samples gathered during the day. In contrast to the control samples, the mRNA expression of the *mdm2* genes increased significantly in samples from patients with psoriasis at three time points (Figure 4A). Moreover, elevated *cop1* mRNA levels were identified diurnally in the non-lesional samples compared to healthy skin samples (Figure 4A). Rhythmic expression of both regulators was not detected in either the control or lesional skin. However, the daily mRNA expression levels of the *cop1* differ significantly in the non-lesional psoriatic skin. In non-lesional skin, the *mdm2* mRNA level peaked at 06:00, whereas for *cop1*, the highest mRNA expression levels were observed at 06:00 and 00:00 (Figure 4B). COP1 was not visible in the cytoplasm, whereas robust nuclear immunostaining was observed in all layers of the normal and non-lesional skin. In contrast, COP1 was detected in the cytoplasm of the lesional epidermis. Differences were not detected in the MDM2 protein expression patterns of the three sample types (Figure 4C).

## 3. Discussion

The relationship between the circadian system and inflammatory diseases has been indicated by several studies [23]. However, the role of the clock genes in the pathomechanism of inflammation is not fully understood. Psoriasis is a chronic inflammatory skin disease characterized by abnormal differentiation and proliferation of the keratinocytes. Therefore, the impact of the circadian regulators should also be examined in psoriatic inflammation and hyperproliferation. To the best of our knowledge, the clock and clock-related genes are poorly investigated in psoriasis. In this study, we report on alterations of the circadian system in psoriasis, with particular regard to asymptomatic skin.

BMAL1, CRY, RORα and REV-ERBα proteins have been described as positive regulators of the anti-inflammatory responses [24,25,26,27,28,29,30,31]. Hence, the downregulation of these proteins or genes can enhance the inflammation. In contrast, CLOCK protein can promote the inflammatory responses via the NF-κB pathway [32]. Our data on human skin samples indicate that the expression levels of *clock*, *bmal1*, *per1* and *per2* circadian genes are significantly higher in psoriatic skin compared to healthy skin. On the other hand, *rev-erbα* was downregulated in the lesional and adjacent uninvolved skin from psoriatic patients compared to healthy skin. The *cry1* transcript was upregulated in uninvolved skin but reduced in psoriatic lesional areas. Moreover, the malfunction of the rhythm of these genes can cause pro-inflammatory processes, which also can be seen in non-lesional psoriatic skin. Similar to our results, a recent transcriptomic study showed that several clock genes (*cry1/2*, *rev-erbα*, *clock*, *bmal1*, *rorα/γ*) with overrepresented binding sites were downregulated in keratinocytes from psoriatic lesions [33]. As the proteins encoded by *cry* genes may have an anti-inflammatory role, it is hypothesized that they may exert this role in non-lesional skin through their upregulation. Furthermore, the different microenvironments (non-lesional vs. lesional) may contribute to the different gene expression patterns that were observed in our study.

Several reports indicated an alteration in clock gene expression in inflammation, i.e., bowel diseases [34,35], arthritis [29,36,37], pulmonary inflammation [38] and numerous types of cancer [39]. Depending on the inflammatory disease, a decrease or increase in the mRNA expression of certain clock genes was found. Disturbance of the daily rhythm of clock genes and abnormal expression of *bmal*, *clock/npas2* and *cry1* were revealed in rheumatoid arthritis [40,41]. Depending on the cancer type, heterogeneous differences in clock gene expression have been described. Downregulated *bmal1* expression was associated with tumor progression in melanoma [42] and breast cancer [43]. A decreased *rev-erbα* transcript level was associated with poor response in gastric cancer [44], whereas enhanced *rev-erbβ* expression correlated with poor overall survival [45]. Low levels of the *per* and *cry* genes were detected in melanoma [46] and head and squamous cell carcinoma [47]. Similar to our findings, in a previous study, downregulation of *rev-erbα* was found in psoriatic lesional and non-lesional skin by high-throughput transcriptome analysis [48]. In accordance with Greenberg et al. [16], we found decreased expression in asymptomatic psoriatic skin compared to the lesional skin, except for the *clock* and *bmal1* genes, whose mRNA levels differed in a daily manner between our samples.

The cell cycle and circadian system consist of bidirectional coupling. Clock genes and the cell cycle regulators can conversely influence the upregulation of their expression, thereby participating in abnormal cell proliferation. Hence, the enhanced clock gene expression may contribute to hyperproliferation in psoriasis.

In parallel with the human ex vivo study, we conducted cytokine stimulation tests to test the pro-inflammatory effects on the core clock genes in HaCaT cells. Pro-inflammatory cytokines are able to decrease the *rev-erbα* mRNA level, which is consistent with our human studies. Liu et al. showed that the transcript level of *rev-erbα* in fibroblast-like synoviocytes was attenuated by TNFα, IL-1β and IL-17. Furthermore, treatment with REV-ERBα agonist SR9009 could suppress the inflammatory effects in rheumatoid arthritis [30]. SR9009 can repress the γδT17 cells and attenuate the inflammatory symptoms, even when applied topically in the psoriasiform model [49]. REV-ERBs are negative regulators of BMAL1; hence, the repression of *rev-erbα* can positively affect the expression of *bmal1*, as can be seen with the TNFα modulation (Figure 3G) [50]. PER1 and PER2 proteins may possess an anti-inflammatory and pro-inflammatory influence, respectively, by controlling the BMAL1 activity [51,52]. *Per2* showed overexpression while the per1 mRNA level decreased in the TNFα group.

Disruption of the circadian system may lead to metabolic syndromes such as type 2 diabetes and obesity, which are also risk factors for psoriasis. In patients with psoriasis, free fatty acids profiles are shifted toward saturated fatty acids [48,53]. Clock proteins, mainly REV-ERBα, have been shown to regulate glucose and lipid metabolism, thereby playing a crucial role in metabolic malfunctions. Downregulation of *rev-erbα* can increase glucose plasma, liver triglyceride and fatty acids levels [54,55], while treatment with REV-ERB agonists or ligands can decrease obesity by reducing the levels of plasma glucose, triglycerides and free fatty acids [56,57]. This may increase the possibility that resettling *rev-erbα* levels could lead to a regression of skin symptoms and could improve the accompanying metabolic syndrome frequently seen in psoriatic patients.

Besides understanding the processes of the bidirectional interaction between several diseases and the circadian clock, the chronotherapeutic approach should be mentioned. It is a treatment approach whereby the goal is to optimize the efficacy of treatment and minimize the adverse effects by taking into account the rhythm of the disease and the bioavailability of the treatment. As reviewed in recent articles, the timing and individually tailoring of medications (by adjusting to the person’s inner clock) is becoming increasingly important in inflammatory diseases or in cancer treatment [58,59,60].

We identified differing epidermal distribution levels of BMAL1, CLOCK, PER2, CRY1 and REV-ERBα proteins via immunohistochemistry staining in healthy vs. psoriatic skin samples. The molecular links between the circadian oscillators and cell division regulatory processes [61] might explain the differential expression levels of clock proteins in the layers of the epidermis in the hyperproliferative psoriatic skin. We found that the REV-ERBα location showed different patterns in the intracellular layers and among the epidermis layers in psoriasis compared to the normal skin. As we know from experimental and in silico studies, *rev-erbα* is upregulated during mitosis [62,63]. The increased cell proliferation in psoriasis may explain the appearance of REV-ERBα in the basal cells and nuclear positivity in all layers of the epidermis. Interestingly, these expression patterns also appeared in the non-lesional psoriatic skin samples. In psoriatic lesions, CLOCK and PER2 immunoreactivity was shown, especially in the granular layer, which was not observed in healthy and asymptomatic psoriatic biopsies. CLOCK/BMAL1 protein complex and PER2 regulate cell cycle checkpoints, whose overexpression is involved in epidermal turnover [64]. In psoriasis, altered distributions can be seen in cell cycle regulatory proteins, CDK2 and cyclin E expression, which cause hyperproliferation and are upregulated in the upper layers of the epidermis [65,66].

PER and CRY have been described as negative regulators in the circadian system. These proteins inhibit the function of E3 ubiquitin ligases MDM2 and COP1, respectively. PER2 prevents the MDM2-mediated ubiquitination of p53 [21], which is essential for the regulation of cell proliferation and is upregulated in psoriasis [65,67,68]. In mammals, CRY proteins might indirectly repress COP1 and its downstream signaling pathways, including the glucocorticoid receptor transcriptional activity [22], which is known as an anti-inflammatory and anti-proliferative mediator [69]. We detected a significant increase in *mdm2* and *cop1* mRNA expression in psoriatic skin compared to normal samples, with no difference in protein expression levels in the epidermis. In future studies, we will plan to examine the alteration of the PER2-MDM2-p53 and CRY-COP1 axes in psoriasis.

In summary, we examined the clock gene expression levels in patients with psoriasis and in pro-inflammatory conditions in vitro. Perturbation of clock gene transcripts was observed in uninvolved and lesional psoriatic areas compared to healthy skin. These findings further show that differences exist between lesional and non-lesional psoriatic skin in terms of histology and inflammatory cell signature. The alterations in clock genes expression among the different skin samples may serve as possible therapeutic (chronotherapy) targets in the future. Modulating the circadian clock via systemic or local therapies (e.g., REV-ERB agonists or ligands) might be effective in not only controlling psoriasis, but also controlling the accompanying inflammatory state.

## 4. Materials and Methods

### 4.1. Study Population

Twelve individuals were enrolled in the study at the Department of Dermatology, University of Pécs, including moderate-to-severe, chronic and plaque-type psoriasis patients (*n* = 6) and healthy controls (*n* = 6). People who work nightshifts and received systemic treatment for their psoriasis or topical steroid creams 2 weeks prior to the biopsy were excluded from the study.

The study protocol was approved and controlled with the written consent of the Regional Research Ethics Committee of the Medical Center, Pécs, with the approval number 64/2008-3280/a. Informed consent was obtained from all subjects and the experiments were carried out according to the ethical principles originating from the Declaration of Helsinki and with Good Clinical Practice as defined by the International Conference on Harmonization.

### 4.2. Skin Biopsies

The human keratinocyte cell line HaCaT (Ams Biotechnology, Ltd., Abingdon, UK) was seeded into 12-well plates in Eagle’s minimal essential medium (EMEM, Lonza, Walkerswill, MD, USA) supplemented with 10% FBS (Gibco, Waltham, MA, USA) and 1% antibiotics. Synchronization of the molecular clock was achieved via serum starvation for 24 h at 70% confluence. After synchronization, cells were stimulated with 50 ng/mL TNFα, IL-22, IL-1b or IL17-A (PeproTech EC, Ltd., London, UK) in EMEM containing 2% FBS. After the 24 h pre-treatment, cells were harvested at 4 h intervals for up to 48 h for RNA extraction.

### 4.3. Quantitative RT-P

Total RNA was extracted using the Direct-zol RNA Miniprep Kit (Zymo Research, Irvine, CA, USA). Reverse transcription was performed from 500 ng of total RNA using SuperScript RT enzyme with random hexamer primers (Thermo Scientific, Waltham, MA, USA). Expression levels of bmal1 (TaqMan assay ID: Hs00154147_m1, Thermo Scientific), per1 (Hs00242988_m1), per2 (Hs00256143_m1), rev-erbα (Hs00253876_m1), cry1 (Hs00172734_m1), clock (Hs04546767_m1), cop1 (Hs01075834_m1) and mdm2 (Hs00234752_m1) were determined by real-time quantitative PCR on a StepOne Real-Time PCR System (Applied Biosystems, Waltham, MA, USA). Gene expression levels were calculated via normalization relative to hprt1 (Hs99999909_m1) or β-actin (Hs99999903_m1) mRNA levels. The _ΔΔ_CT method was used to quantify differences in clock gene mRNA contents.

### 4.4. Immunohistochemistry

Immunohistochemistry was performed as previously described using the following primary antibodies: anti-CLOCK (abcam; ab65033), anti-PER2 (abcam; ab227727), anti-CRY1 (Santa Cruz; sc-101006), anti-REV-ERBα (Origene; TA350237), anti-COP1 (Santa Cruz; sc-166799), anti-MDM2 (Origene; AP08104PU-N) [46]. The sections were counterstained with hematoxylin. The staining was scored using the H-score method based on the nuclear staining intensity (0, 1+, 2+, 3+) and calculated with the formula [1× (% cells 1+) + (2× (% cells 2+) + (3× (% cells 3+)].

### 4.5. Data Analysis

The statistical analysis was performed using GraphPad Prism v8. The significance of differences between values was determined by either Student’s t-test, Mann–Whitney t-test or two-way ANOVA. Differences with *p* < 0.05 were considered statistically significant.

## Figures and Tables

**Figure 1 ijms-23-00121-f001:**
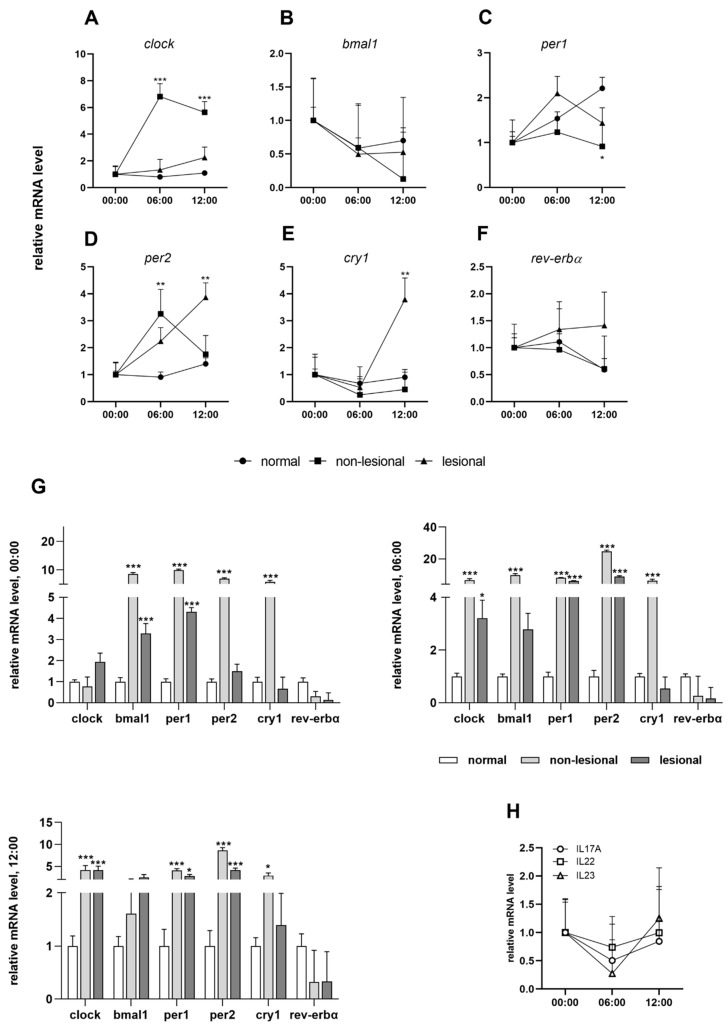
Circadian entrainment in psoriasis: (**A**–**F**) daily patterns of clock genes in normal, psoriatic non-lesional and lesional skin samples; (**G**) relative mRNA expression levels of the indicated genes at 12:00, 06:00 and at 00:00; (**H**) daily patterns of IL-17A, IL-22 and IL-23 mRNA expression. Significant differences assessed by two-way ANOVA followed by Turkey’s post hoc test between the time points: * *p* < 0.05; ** *p* < 0.005; *** *p* < 0.0005. Data are presented as the means ± SEM.

**Figure 2 ijms-23-00121-f002:**
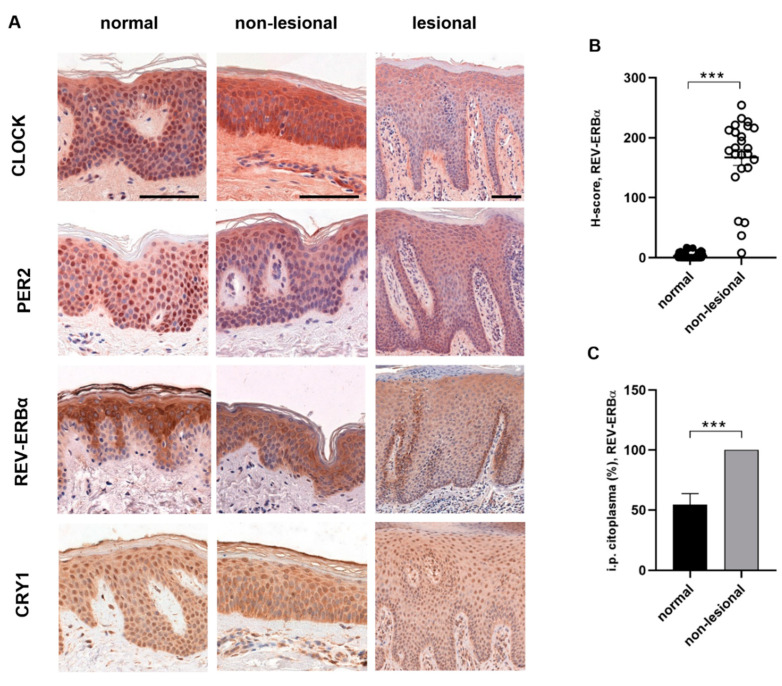
Immunohistochemistry staining of circadian proteins in human skin: (**A**) representative DAB-stained skin sections at 06:00. Normal and non-lesional skin samples at 400× magnification (scale bar = 100 µm) Lesional skin at 200× magnification (scale bar = 100 µm). (**B**) H-scores (the percentages of cells at each staining intensity level) for nuclear REV-ERBα expression in normal and non-lesional epidermis: *** *p* < 0.0005. Data are presented as the means ± SEM. (**C**) Cytoplasmic staining (ratio of the positive cells) of REV-ERBα in normal and non-lesional epidermis: *** *p* < 0.0005. Data are presented as the means ± SD.

**Figure 3 ijms-23-00121-f003:**
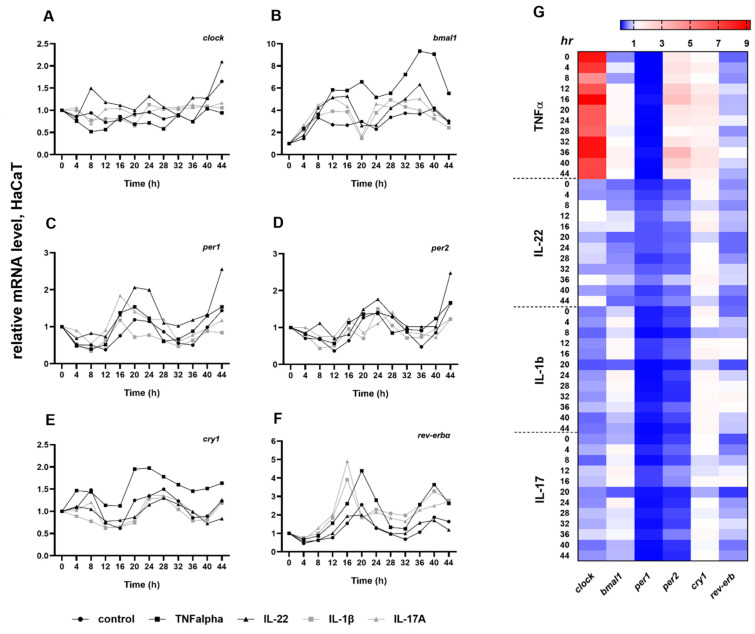
Circadian entrainment in HaCaT keratinocytes. (**A**–**F**) The effects of cytokines on the circadian entrainment in HaCaT keratinocytes. Data are plotted as means only. The statistical significance of differences was analyzed by two-way ANOVA followed by Dunnett’s post hoc test. The results of two-way ANOVA are described in Appendix A. (**G**) Heatmap expression plot of selected clock genes. Rows represent time points and columns represent genes.

**Figure 4 ijms-23-00121-f004:**
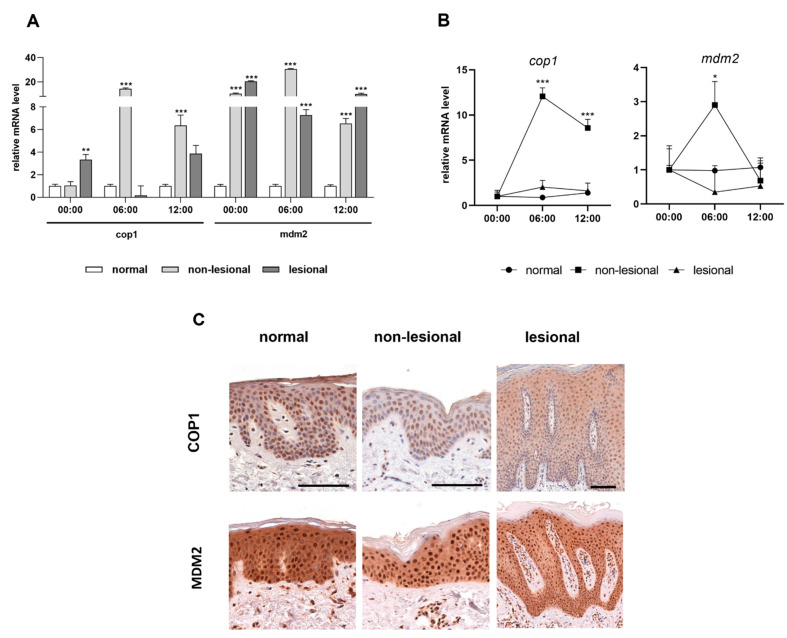
Expression levels of the cell cycle regulators in psoriasis**.** (**A**) Relative mRNA expression of *cop1* (**left**) and *mdm2* (**right**) genes at three time points. (**B**) Daily changes of mRNA expression of *cop1* (**left**) and *mdm2* (**right**) genes. Significant differences assessed by two-way ANOVA followed by Turkey’s post hoc test between the time points: * *p* < 0.05; ** *p* < 0.005; *** *p* <0.0005. Data are presented as the means ± SEM. (**C**) Representative DAB-stained skin sections at 06:00.

## Data Availability

All original data are available from the corresponding author upon request.

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
