# Peer review of "Expression Patterns of Clock Gene mRNAs and Clock Proteins in Human Psoriatic Skin Samples"

_ijms, 2021, doi:10.3390/ijms23010121_

Round 1
Reviewer 1 Report
I read with great interest this manuscript titled "Expression patterns of clock gene mRNAs and clock proteins in human psoriatic skin samples" by Németh.
ABSTRACT
Please specify in the introduction that psoriasis is a systemic inflammatory disease that negatively modulate both sleep and circadian rhythm.
Please clearly mention the chronomeidcal approach
INTRODUCTION
Line 46: psoriasis is also Th1 and Th22 mediated.
Line 47: Please cite directly the new Global Burden of Diseases of psoriasis worldwide: Damiani G, Bragazzi NL, Aksut CK, et al. The Global, Regional And National Burden Of Psoriasis: Results And Insights From The Global Burden Of Disease 2019 Study. Front Med (Lausanne) 2021; 8:743180. doi: 10.3389/fmed.2021.743180
DISCUSSION
Please add in the discussion the approach of chronomedicine falls in the precision medicine and in the temptative to increase therapy response minimizing adverse events [10.3390/jcm9010186]
Reviewer 2 Report
the authors make an important connection between circadian clock genes, keratinocytes and psoriasis. They need to do more work on it:
1- I would like to know if the authors tried to perform ELISA for protein that they detect by IHC..is that possible? I know that some company sell ELISA kit for clock protein...if so it would be interesting and the authors could also divided epidermal from dermal.
2- It would be interesting to see in an invite model if they are able to control psoriasis blocking or enhancing cytokines in the skin during the day applying topically? Could the authors perform this type of experiment? If not could you explain why?
3- Maybe I missed this point but could you discuss your results on psoriasis in relation at other inflammatory disease as cancer?
4- minor point but still important: check carefully your typo error and some English error.
Reviewer 3 Report
This is an intriguing study. Using qRT-PCR, IHC and cell culture authors have shown expression pattern of circadian rhythm genes in psoriasis. These data show induction of some of the genes in non-lesional psoriasis skin. This is interesting and could mark pathogenic changes in non-lesional skin crucial for psoriasis manifestation. Manuscript is nicely written and experiments are well performed.
Major concerns,
- IHC analysis for tested genes (by qRT-PCR) does not confirm their regulation on protein level (Figure 2). What could be a reason for this?
- Although, authors claim that expression pattern of REV-ERBα was different among healthy and non-lesional skin, the signal intensity seems higher in normal skin, suggesting normal skin has more REV-ERBα expression (Figure 2A). If this is image specific; replace it with a better representing image. Also provide scoring criteria in figure 2B-C.
- For figure 3, TNF was able to induce circadian rhythm genes in cultured HaCaT cells. What is the mechanism of this?
- Anti-TNFs are successful in treating psoriasis to some extent. Could there be a link between TNF pathway in keratinocytes and circadian rhythm in psoriasis.
- As in psoriasis, there are multiple cytokines present at the same time in the inflammatory milieu. Have authors tried treating HaCaT cells with cytokine mix (for example, TNF+IL-17+IL-22) and judged clock gene expression.
- A recent transcriptomic study in psoriasis shows that circadian rhythm pathway was the top enriched pathway among downregulated genes (for example cry1 and 2) in psoriatic keratinocytes. In comparison to this what authors thinks about the upregulation of circadian rhythm pathway genes in non-lesional skin. This article should be cited and discussed in the discussion section.(PMID: 30320872)
Minor concerns,
- What is the rationale to choose cell cycle genes to study here? Include a short section in results describing how these genes are linked to circadian rhythm.
- Figure 1A-F, mark the Y axis is it relative/normalized expression or mRNA fold change.
- Figure 3A-F, mark the Y axis is it relative/normalized expression or mRNA fold change.
Round 2
Reviewer 2 Report
Thanks to the authors for their work
Reviewer 3 Report
This is an fantastic study highlighting circadian rhythm pathway in psoriasis. The authors have responded to all the raised concerns and provided an adequate response. The new added sections adds to the core message of the story and enhances the quality of the article significantly.